# Acute Effect of a Saffron Extract (Safr’Inside^TM^) and Its Main Volatile Compound on the Stress Response in Healthy Young Men: A Randomized, Double Blind, Placebo-Controlled, Crossover Study

**DOI:** 10.3390/nu15132921

**Published:** 2023-06-27

**Authors:** Camille Pouchieu, Line Pourtau, Julie Brossaud, David Gaudout, Jean-Benoit Corcuff, Lucile Capuron, Nathalie Castanon, Pierre Philip

**Affiliations:** 1Activ’Inside, F-33750 Beychac et Caillau, France; l.pourtau@activinside.com (L.P.); d.gaudout@activinside.com (D.G.); 2Hormone Laboratory, Nuclear Medicine, CHU Bordeaux, UMR INRA 1286, University Bordeaux, F-33000 Bordeaux, France; julie.brossaud@chu-bordeaux.fr (J.B.); jean-benoit.corcuff@chu-bordeaux.fr (J.-B.C.); 3INRAE, Nutrition and Integrative Neurobiology (NutriNeuro), Bordeaux University, UMR 1286, F-33000 Bordeaux, France; lucile.capuron@inra.fr (L.C.); nathalie.castanon@inrae.fr (N.C.); 4Pôle Neurosciences Cliniques, Centre Hospitalier Universitaire de Bordeaux, F-33076 Bordeaux, France; pierre.philip@u-bordeaux.fr; 5Sommeil, Addiction et NeuroPSYchiatrie, Bordeaux University, CNRS, SANPSY, UMR 6033, F-33000 Bordeaux, France; 6Centre d’Investigation Clinique Bordeaux, INSERM CIC 1401, Centre Hospitalier Universitaire de Bordeaux, F-33000 Bordeaux, France

**Keywords:** saffron extract, safranal, terpene, stress, anxiety, corticotropic axis

## Abstract

According to animal studies, saffron and its main volatile compound safranal may reduce biological and behavioral signs of acute stress. However, little is known about its impact in humans. This study investigated the acute effect of a saffron extract and safranal on the biological and psychological stress responses in healthy men experiencing a laboratory stress procedure. In this double-blind, placebo-controlled, randomized, cross-over study, 19 volunteers aged 18–25 received a single dose of 30 mg saffron extract (Safr’Inside^TM)^, 0.06 mg synthetic safranal, or a placebo on three visits separated by a 28-day washout. Thirteen minutes after administration, participants were exposed to the Maastricht acute stress test (MAST). Salivary cortisol and cortisone were collected from 15 min before the MAST (and pre-dose), 3 min before the MAST, and then 15, 30, 45, 60, and 75 min after the MAST, and stress and anxiety were measured using visual analogic scales. Compared to the placebo, stress and anxiety were significantly toned down after Safranal and Safr’Inside^TM^ administration and coupled with a delay in the times to peak salivary cortisol and cortisone concentrations (*p* < 0.05). Safr’Inside^TM^ and its volatile compound seem to improve psychological stress response in healthy men after exposure to a lab-based stressor and may modulate the biological stress response.

## 1. Introduction

Stress may be defined as “a particular relationship between the person and the environment that is appraised by the person as taxing or exceeding their resources and endangering his or her wellbeing” [1]. Exposure to stress is known to trigger a complex interplay of nervous, endocrine, and immune mechanisms [2]. Playing a central role, the hypothalamus–pituitary–adrenal (HPA) axis is activated, resulting in an adaptative increased cortisol secretion. The latter was investigated via salivary cortisol secretion peaks between 20 and 40 min after exposure to the stressful event [3]. Conversely, if this exposure is actually or perceived as intense, repetitive, or prolonged, the stress response may become maladaptive and may induce deleterious cognitive, emotional, physical, and behavioral symptoms [4].

In modern life, people may be exposed daily to different sources of stress, such as constant digital connection, heavy workload, changes in sleep wake schedules, or fear of climate change. It has been shown that chronic stress is associated with an increase in the global prevalence of mental and physical health problems, such as anxiety and depression [5,6,7], but also cardiovascular diseases [8,9]. In addition, it is now known that the COVID-19 pandemic-related stress has affected the brain and mental health. Indeed, COVID-19 has led to a 27.6% increase in cases of major depressive disorder and a 25.6% increase in cases of anxiety disorders worldwide in 2020 [10]. Becoming aware of the side effects from conventional anxiolytics such as benzodiazepines or other psychotropic drugs, people are seeking more safe and natural alternatives in order to cope with the adverse effects of stress, such as yoga or dietary supplements [11,12]. Indeed, the European stress management supplements market is expected to grow at a compound annual growth rate of 2.3% through the 2017–2030 period [13]. This “over-the counter” medication may be effective in the management of psychological and physiological stress responses, yet broad evidence based on clinical trials or experimental data is lacking [14]. Combining pharmacological and non-pharmacological interventions, such as nutraceuticals, phytoceuticals, or psychological interventions, to manage stress-related problems may be the most effective approach, taking into account individual patients’ characteristics [15,16].

Among natural alternatives, saffron extract produced from the dried stigma of *Crocus sativus* L. appears to be a promising candidate. Saffron stigma contains volatile compounds, mainly terpenes such as safranal, as well as non-volatile ones such as crocins, crocetins, picrocrocins, and flavonoids [17]. Interestingly, saffron extract may reduce the plasma levels of corticosterone in response to acute stress in rodent studies. This suggests a potential modulation of the HPA axis following a single dose of such an extract but the effect of its isolated constituents, i.e., crocins and safranal, are inconsistent [18,19,20]. To our knowledge, only two clinical studies have investigated the biological and psychological anti-stress effect of saffron in chronically stressed women. The first demonstrated that women affected by menstrual distress and who inhaled saffron for 20 min experienced significant decreased salivary cortisol coupled with reduced self-perceived anxiety compared to those receiving the placebo, suggesting a potential anti-stress and anxiolytic effect likely induced by the volatile compound safranal [21]. Indeed, the means of salivary cortisol was about 0.11 ug/dl after saffron inhalation compared to 0.18 ug/dl in the placebo group (*p* < 0.0051). The second clinical study showed a beneficial effect of a single dose of Safr’Inside^TM^ on another biological response to stress, namely heart rate variability (root mean square of the successive differences (RMSSD)), after an induced laboratory acute stressor (the observed multitasking stressor) in healthy adults experiencing low mood [22]. Indeed, participants who received placebo had a decreased RMSSD during the laboratory stressor, whereas participants receiving Safr’Inside^TM^ experienced no change in RMSSD (*p* = 0.003). Although the effect of saffron on the HPA axis was assessed, the participants had anticipated the stressor, rendering the results uninterpretable. Otherwise, Safr’Inside^TM^ was well tolerated and associated with reduced depression scores and improved social relationships after 8 weeks of supplementation.

Therefore, the aim of the present study was to investigate the acute effect of a saffron extract and its isolated main volatile compound tested as synthetic safranal on the biological and psychological responses induced by an acute validated laboratory stress procedure in healthy young men.

## 2. Materials and Methods

### 2.1. Participants

Twenty healthy men, aged 18–25 years, with a body mass index between 18.5 and 25.0 kg/m^2^, who had not smoked since at least 3 months, with usual weekdays waking hours 6.00–9.00 am, were recruited between September 2020 and December 2021 at the SANPSY Unit located in the Teaching University Hospital, Bordeaux, France (see Figure 1 for the Consolidated Standards of Reporting Trials (CONSORT) diagram). The participants were recruited using the investigational center’s participant database and the study was advertised by local newspapers and social networks. Participants with high blood pressure (BP > 140/90 mmHg); a history of or currently suffering from mental disorders, infectious diseases, diabetes mellitus, allergic diseases, cardiovascular diseases, unbalance thyroid diseases; or subjects taking anxiolytic, antidepressants, or any other treatment likely to affect salivary cortisol measures (such as steroids, including ointments) were not eligible. In addition, exclusion criteria likely to bias salivary cortisol measures were: the occurrence of a life event 2 weeks before enrollment or a planned life event during the study period (death, divorce, change in professional activities, surgery, travel involving jet lag, etc.), shift workers or those working in extreme conditions (in cold rooms), those engaged in high level of physical activities, those who regularly consumed >3 coffee cups in the morning, those who regularly consumed >3 alcoholic drinks per day, those who used dietary supplements in the 2 weeks before enrolment, and those using illicit drugs. In order to enroll participants likely to experience a biological stress response after the acute stress protocol [23], only subjects displaying an increase in salivary cortisol from 15 to 30 min post-stressor greater than 2× the coefficient of variation (i.e., >16%) of the cortisol assays at the screening visit were eligible.

### 2.2. Tested Products

Participants were randomly allocated to receive a single dose of 30 mg of a full-spectrum standardized saffron extract (Safr’Inside^TM^, Activ’Inside, Beychac et Caillau, France) obtained according to the patent EP3490575B1-WO2017EP69200, which contains crocins (mainly trans-4-GG, trans-3-Gg; cis-4-GG, trans-2-G) > 3%, safranal > 0.2%, picrocrocin derivatives (mainly picrocrocin, HTCC) > 1%, and kaempferol derivatives (mainly kaempferol-3-sophoroside-7-glucoside, kaempferol-3-sophoroside) > 0.1%, analyzed using the U-HPLC method; 0.06 mg synthetic safranal (≥90% stabilized, Sigma-Aldrich, Darmstadt, Germany), corresponding to the dose provided by 30 mg Safr’Inside^TM^; or a placebo containing maltodextrin. Each product (saffron extract, synthetic safranal, or placebo) was presented as a single hard-shell capsule with same appearance (color, size, odor) containing 100 mg of powder and was administered sublingually during the testing visits (V1, V2, and V3). Each capsule was provided on the visit day in opaque pill dispensers.

### 2.3. Study Design and Protocol

This study adopted a randomized, double-blind, placebo-controlled, cross-over (3 arms) design. Each participant went through one screening visit (V0) followed by three testing visits (V1, V2, and V3), during which they consumed either the saffron extract, the safranal or the placebo capsule. Following criteria checks for inclusion (V0), eligible participants provided lifestyle and demographic data, and their weight, height, heart rate, and blood pressure were measured. Participants were asked not to change their dietary, sleeping, and exercise habits throughout the whole study. In addition, they were requested not to consume saffron-based food or beverages, alcohol, and not to practice moderate to vigorous physical activities within the 24 h preceding a testing visit. Then, they performed a training session on the MAST. Saliva samples were collected before and immediately after the MAST and every 15 min for 1 h. At the end of the visit, participants were provided with sleep and food diaries and study instructions. Participants that had no reactive cortisol response 15 min after the MAST were not eligible.

The V1 visit was planned 21 +/− 2 days after the inclusion visit, while the V2 and V3 visit were planned following a washout period (28 +/− 2 days). During each testing visit, participants arrived at the research unit at an agreed time in early afternoon that remained consistent across all testing visits. A standardized lunch free from saffron, dark chocolate, caffeine, herbs, and alcohol was provided. Sleep and food diaries were checked. The absence of recent consumption of cannabis and alcohol were confirmed via a urine tetrahydrocannabinol test and a breath alcohol test. The assessment timeline during study visits is shown in Appendix A. One hour after a standardized lunch, participants rinsed their mouth 15 min before the collection of a saliva sample and the rating of their self-perceived stress and anxiety using visual analogic scales (VAS). Stress and anxiety were rated by making a vertical mark on a 100 mm straight horizontal line orientated from the left (“not at all”) to the right (“very much”). Participants were randomly assigned to a treatment sequence order. Randomization was based on a computer-generated list provided by staff not involved in the clinical assessment or in the data analyses. The randomization was equilibrated by block size of 6. The capsule of the tested product was opened by the participant and totally poured under his tongue. Saliva samples and self-perceived stress and anxiety levels were obtained 15 min after product intake. The MAST was administered to the participants. Finally, saliva samples and self-perceived stress and anxiety were collected after the end of the MAST and every 15 min for 1 h.

### 2.4. Maastricht Acute Stress Test

The MAST is a validated experimental stress procedure that has been widely used as a safe and reliable measure to activate both acute physiological and subjective measures of the stress response in healthy adults [24,25]. A 5 min preparation phase allowed the participant to read the instructions for the upcoming tasks (on a PowerPoint presentation). During the 10 min acute stress phase, participants were required to place their hand in cold water at 2 °C and complete an arithmetic task (counting backward by 13, 17, 19, or 23) for alternating time intervals. The sequence (order and duration) of the various hand immersion trials and mental arithmetic tasks was the same for all participants but differed according to visit.

### 2.5. Salivary Cortisol and Cortisone

Salivary samples were obtained using a Salivette kit (Sarstedt, Nümbrecht, Germany) to measure cortisol and cortisone concentrations. Once collected, samples were stored at 4 °C before being analyzed. Salivary cortisol and cortisone were measured by LC-MS/MS (Prominence liquid chromatography system; Shimadzu, Nakagyo, Japan; and 5500 Qtrap detector, Sciex, Framingham, MA, USA). A liquefying agent (Sputasol; Thermo Fisher Scientific, Waltham, MA, USA) was added to 400 µL of saliva sample, which was then incubated for 30 min at 37 °C. Next, solid-phase extraction with a hydrophilic lipophilic balance (Waters^TM^) was performed before injecting the extract into the LC-MS/MS system. The coefficients of variability for the cortisol and cortisone assays were 8 and 9%, respectively.

### 2.6. Statistical Analyses

As no clinical study has investigated the effect of saffron extract in response to an acute laboratory stressor on salivary cortisol response, the sample size calculation was based on the effect size of a lemon balm food supplement on salivary cortisol in healthy adults 1 h post-treatment [26]. A sample size of 18 subjects was hypothesized to detect an effect size of 1.4 on salivary cortisol response, with a two-sided 5% significance level and a power of 80%. Considering a 10% drop-out rate, 20 volunteers were included in order to obtain 18 completions. Among the 20 included participants, 1 participant withdrew before randomization and 1 participant discontinued the study after visit 1, leading to missing data at visit 2 and 3 for all biological and psychological parameters.

Demographic and lifestyle characteristics at baseline (V0) and pre-treatment values of VAS stress, VAS anxiety, salivary cortisol and cortisone collected during testing visits were compared according to the sequence using a linear mixed model (PROC MIXED). The model included sequence, period, and treatment as fixed effects and participants within sequence as the random effects. In order to confirm that participants found the MAST stressful, VAS anxiety, VAS stress, salivary cortisol, and cortisone were compared before, during, and after the MAST using the linear mixed model as above and using the time of sample collection as additional fixed factor.

The incremental area under the curve (AUC) for VAS stress, VAS anxiety, salivary cortisol, and cortisone were calculated using the trapezoid rule, discarding the area beneath the pre-treatment values, considered the baseline value [27]. As AUC, maximum concentration (Cmax) and time to reach Cmax (Tmax) did not exhibit the normal distribution of residuals regardless of transformation, the Friedman ANOVA test was used to analyze the treatment effect with post hoc analyses using Tukey’s method when appropriate. Due to biological non-response to the stressor, 5 datapoints were judged to be abnormal regarding Tmax and Cmax and were excluded from these analyses.

All statistical tests were 2-sided, and a *p*-value < 0.05 was considered statistically significant. All data were analyzed using SAS version 9.4 (SAS Institute, Cary, NC, USA). Figures were computed with GraphPad Prims (version 9.4.0, GraphPad Software, San Diego, CA, USA).

## 3. Results

### 3.1. Inclusion and Population Characteristics

The study population involved 19 healthy males with a mean age of 22.4 ± 2.0 years. All participants reported being healthy, not consuming any food supplements, medications, or illicit drugs that would interfere with the test product. All baseline values were within the physiological range (Table 1). The participants usually consumed less than an average of one cup of coffee and alcoholic drink per day. Participants displayed a 94% (±59) mean increase in salivary cortisol from 15 to 30 min after the end of the MAST, showing a reactive cortisol response. Test capsules were well tolerated, and no adverse effects were reported during the study. Except for one participant that dropped out after the 1st test visit without giving any reason, all subjects completed the trial.

No statistically significant differences were observed at baseline among the six sequence groups with respect to demographic, lifestyle, or psychological and physiological characteristics. Therefore, randomization was considered successful.

### 3.2. Validation of the MAST

The analyses of the VAS revealed that the participants, irrespective of treatments, experienced a peak of subjective anxiety and stress 15 min after the MAST (Figure 2a,b), which was coupled to a peak of salivary cortisol and cortisone 30 min after the MAST (Figure 2c,d), which is indicative of a psychological and hormonal stress response (effect of time *p* < 0.0001 for all outcomes).

### 3.3. Self-Perceived Anxiety and Stress

With regard to anxiety VAS, there was a significant difference in VAS anxiety response (AUC_t2t4_ under the anxiety peak) between the three treatments (*p* = 0.04, Friedman’s test). The subjective anxiety response was lower in the saffron extract and safranal groups compared to the placebo group (Figure 3a,b). The multiple pairwise comparison showed a significant difference between the safranal and placebo groups (*p* = 0.01, Tukey’s test).

With regard to stress VAS, there was a significant difference in VAS stress response (AUC_t2t4_ under the stress peak) between the three treatments (*p* = 0.03, Friedman’s test). The subjective stress response was lower in the saffron extract and safranal groups compared to the placebo group (Figure 3c,d). The multiple pairwise comparison showed a significant difference between safranal and placebo groups (*p* = 0.01, Tukey’s test).

### 3.4. Salivary Cortisol and Cortisone Concentration

With regard to salivary cortisol, there was a significant treatment effect regarding the time to reach maximal salivary cortisol concentration (T_max_) (*p* = 0.04, Friedman’s test). The participants receiving an acute saffron extract supplementation had a higher Tmax for salivary cortisol compared to placebo in response to the acute stress (*p* = 0.04, Tukey’s test) (Figure 4a). There was a significant treatment effect on salivary cortisone peaking time (T_max_) (*p* = 0.04, Friedman’s test) (Figure 4b). The participants receiving an acute saffron extract supplementation had a longer salivary cortisone Tmax compared to placebo in response to the acute stress (*p* = 0.04, Tukey test) (Figure 4b). No other effects of treatment were observed on salivary cortisol (C_max_, AUC_t1t7_) nor salivary cortisone (C_max_, AUC_t1t7_) (Table 2).

## 4. Discussion

The current study is the first double-blind, randomized, crossover study that explored the effect of a single dose of a proprietary saffron extract (Safr’Inside^TM^) or its main volatile compound safranal, on the biological and psychological stress responses in healthy young men after experiencing a physically and psychologically challenging laboratory stress test (MAST), compared to a placebo group. As expected, both biological and psychological parameters responded to the MAST with stressor-induced changes. The levels of self-perceived stress and anxiety using VAS were toned down after Safr’Inside^TM^ and safranal administration compared to the placebo. In addition, we observed a significant effect of the saffron extract on the glucocorticoid stress response, showing a delay in the salivary cortisol and cortisone responses in participants receiving the saffron extract compared to those receiving the placebo.

Irrespective of the treatment, the participants found the MAST to indeed be stressful, as suggested by the increase in perceived anxiety and stress 15 min after completing the stressor and to a lesser extent in the salivary cortisol and cortisone 30 min post-stressor. As expected, the HPA axis stress response related to cortisol secretion was in line with previous studies using the MAST in healthy young men [24,28]. However, the basal salivary cortisol concentration of these study’s participants was clearly lower than observed in other healthy populations using different acute stress protocols [23,24,28,29]. This may be partly explained by the method of cortisol assays. Indeed, in our study, salivary cortisol was quantified by the method of choice by LC-MS/MS, which has higher sensitivity and specificity than conventional immunoassays used in clinical practices [30]. Thus, other studies using immunoassays to assay salivary cortisol suffered from some degree of cross-reactivity of assays with cortisone, leading to higher apparent values of cortisol. In addition, this low basal level of salivary cortisol strongly suggests that the participants tested here were indeed healthy and relaxed.

To our knowledge, no clinical study has investigated the efficacy of an acute saffron extract or safranal neither using oral nor sublingual administration in an acute experimental stress setting on psychological and biological markers, including salivary cortisol and its metabolite, cortisone. Salivary cortisol is commonly used as a surrogate marker of serum free cortisol in psychophysiological stress research with its tremendous advantage of non-invasive sample collection. Recently, salivary cortisone has been identified as a promising stress marker that showed a high discriminatory power and significant associations with subjective stress measures [31]. Indeed, salivary cortisol is rapidly, locally, and irreversibly converted to cortisone, as the salivary glands exhibit high levels of 11ß-hydroxysteroid dehydrogenase 2 (11ß-HSD2). This results in 2–6 times higher concentrations of salivary cortisone compared to cortisol in saliva, consistent with what is observed in our study.

Interestingly, the saffron extract and safranal increased the peaking time to reach the maximum concentration of salivary cortisol and cortisone after acute stress in comparison to the placebo, suggesting a delayed glucocorticoid secretion. However, the multiple pairwise comparison revealed a significant difference between the saffron extract and placebo groups. The anxiolytic effect of saffron was previously demonstrated in patients with mild to moderate depression [32,33,34], but the chronic supplementation of saffron extracts may not consistently modify feelings of anxiety in healthy individuals self-reporting low mood [22,35]. The delay in glucocorticoid secretion after saffron administration highlighted its modulatory role on the HPA response to an acute stress exposure. Only one randomized clinical study has investigated the efficacy of anti-stress saffron aromatherapy, but the study was conducted in chronically stressed female students suffering from menstrual troubles without including any experimental acute stressors [21], leading comparisons difficult. After 20 min of exposure, saffron odor significantly decreased salivary cortisol in both follicular and luteal phases (by almost 1.7 and 0.9 nmol/L, respectively), compared to ethanol inhalation, and was also associated with reductions in the symptoms of anxiety. As this is a preliminary study conducted in healthy non-stressed men, it will be necessary to confirm the clinical significance of the HPA axis modulation in a larger sample including both sexes suffering from natural “day to day” stress.

It is well known that stress activates the HPA axis, leading to the plasma corticosterone increase as a response in mice [36]. In male mice exposed to electroshock stress, several studies showed that different doses of saffron extracts via intraperitoneal administration prevented elevations in plasma corticosterone but findings regarding the administration of isolated components, such as crocin or safranal, were inconsistent across research [19,20]. Although the biological mechanisms by which saffron delayed the stress-induced glucocorticoid secretion needs further investigations, our findings are supported by mechanistic studies: saffron may inhibit corticosterone secretion in stressed mice by reducing the gene expression of the corticotrophin-releasing hormone in hypothalamus [37] and by blocking NMDA glutamate and/or sigma opioid receptors located in the adrenal cortex [20,38].

Participants receiving Safr’Inside^TM^ or safranal felt less stressed and anxious than those receiving placebo on a short period of 30 min after the stressor onset. The pairwise comparison revealed that these feelings were significantly different between safranal and placebo. As Safr’Inside^TM^ is composed of more than hundreds of metabolites, including safranal, this latter may be subjected to a “matrix effect”, contrary to pure safranal. This matrix effect may accelerate or delay the absorption of safranal, leading to a maximum effect on perceived stress and anxiety either few minutes earlier or later than pure safranal. In addition, as these feelings were not continuously collected, it is likely that the maximum perceived effect occurred in the 15 min period separating two assessments of VAS. Although no clinical study has explored the anxiolytic and anti-stress effect of synthetic safranal, its anxiolytic effect was supported by some preclinical studies [18,19]. However, experimental design may not be comparable due to discrepancies between the tested doses, administration route, and timing of outcome collection. Indeed, Hooshmandi et al. showed that intraperitoneal (i.p.) pretreatment with safranal can inhibit behavioral signs of stress in male Wistar rats, although these signs were explored more than 20 days after electroshock stress induction. In addition, i.p. safranal administration demonstrated anxiolytic activity in a dose-dependent manner in mice experiencing the elevated-plus maze test, the most commonly employed animal behavioral test to measure anxiety-like behavior. As safranal exerts anxiolytic effects similar to those of anxiolytic medications (diazepam), researchers hypothesized that it may act by interacting with the benzodiazepine binding site at the GABA_A_ receptor, although no study has really demonstrated this neurobiological effect so far [18,36].

The primary strength of this study includes the use of both objective and subjective outcomes measures to assess the anti-stress effect of a saffron extract and its main volatile compound, safranal, in healthy non-stressed men experiencing a validated laboratory stress procedure (MAST). Secondly, as the anti-stress effect should be collected quickly after product intake, we administered the tested product via the sublingual route to provide the immediate onset of pharmacological effect [39].

However, despite several positive findings from this study, some limitations should be acknowledged, suggesting directions for future research. Firstly, as greater acute HPA responses have been found in adult men as compared to adult women [40], we investigated the stress response only in healthy young men to increase the likelihood of detecting the anti-stress effect of the tested products. We excluded women due to the influence of sex hormones, which fluctuate with the menstrual cycle, and the influence of oral contraceptives on cortisol levels. The generalizability of our results to other participant demographics, such as women or chronically stressed people, thus remains to be tested in future research. Secondly, while a linear mixed model considering the sequence, period, time of collection, and treatment would have been the statistical method of choice, the restricted sample size in combination with a non-Gaussian distribution of physiological and psychological parameters only allowed for a more basic statistical approach.

## 5. Conclusions

This study reports that a single dose of Safr’Inside^TM^ in healthy young men delayed the typical peak of salivary cortisol and cortisone appearing in response to a physical and psychosocial stressor, in comparison to a placebo. This interesting finding therefore supports the biological role of Safr’Inside^TM^ in the stress management. In addition, we also showed, for the first time in a human study, that Safr’Inside^TM^ and its main volatile compound safranal may reduce the level of perceived stress and anxiety, although the underlying biological mechanism needs further investigations.

## Figures and Tables

**Figure 1 nutrients-15-02921-f001:**
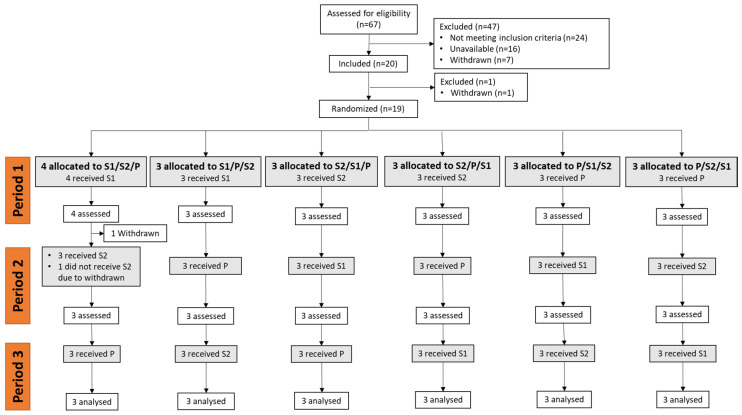
Consolidated Standards of Reporting Trials (CONSORT) flowchart diagram. S1: saffron extract; S2: safranal; P: placebo.

**Figure 2 nutrients-15-02921-f002:**
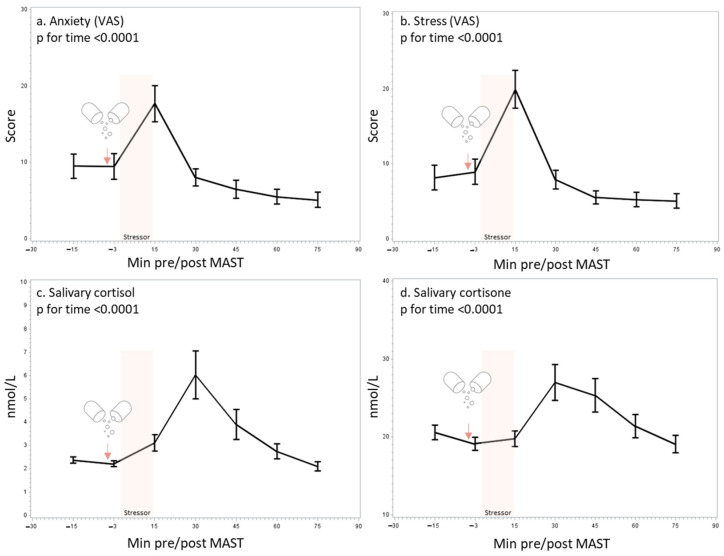
Effect of the Maastricht acute stress test (MAST) on self-perceived anxiety (**a**) and stress (**b**) using a visual analog scale (VAS), salivary cortisol (**c**), and salivary cortisone (**d**). Mean and SEM are shown.

**Figure 3 nutrients-15-02921-f003:**
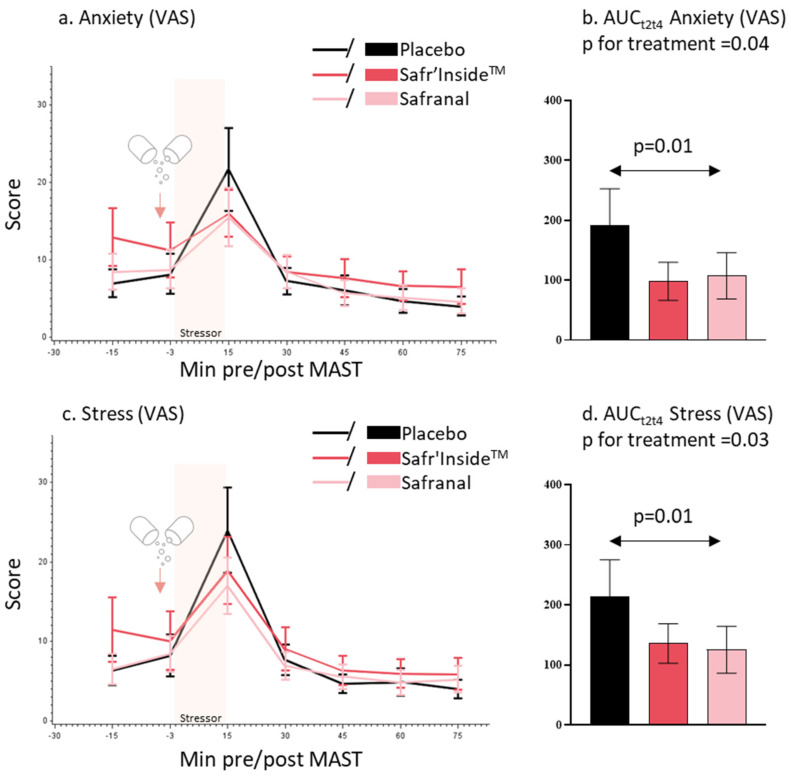
Mean and SEM of self-perceived anxiety (**a**) and stress (**c**) using visual analog scale (VAS) by minutes pre/post MAST and treatment. Mean and SEM of area under the curve (AUC_t2t4_) for VAS anxiety (**b**) and stress (**d**) by treatment.

**Figure 4 nutrients-15-02921-f004:**
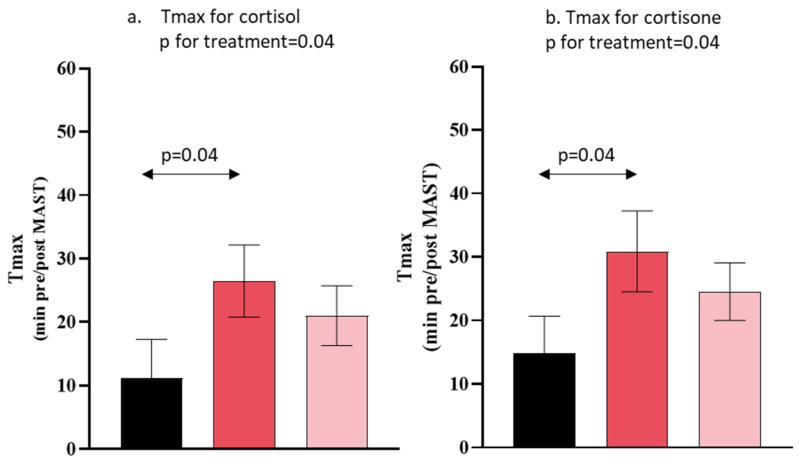
Mean and SEM of Tmax for salivary cortisol and cortisone by treatment. Placebo (in black), Safr’Inside^TM^ (in red) and Safranal (in pink).

**Table 1 nutrients-15-02921-t001:** Baseline characteristics of the population (N = 19).

	Mean	SD
Age (years)	22.4	2.0
Body mass index (kg/m^2^)	22.1	1.6
Systolic blood pressure (mmHG)	113.5	8.0
Diastolic blood pressure (mmHG)	72.8	6.4
Heart rate (bpm)	73.0	14.7
Coffee consumption (units/day)	0.6	0.9
Alcohol consumption (units/day)	0.2	0.5
Normal sleep duration (hours)	8.0	1.0
Salivary cortisol change from 15 to 30 min after the MAST (%)	94	59

**Table 2 nutrients-15-02921-t002:** Salivary cortisol and cortisone parameters according to the treatment group.

	Placebo (*n* = 17)	Safr’Inside^TM^ (*n* = 19)	Safranal (*n* = 18)	*p*-Value
Salivary cortisol				
AUC_t1t7_ (nmol/L*min)	58 (26)	67 (33)	58 (26)	0.17
C_max_ (nmol/L)	6.9 (1.9)	7.4 (2.4)	6.4 (1.6)	0.90
T_max_ (minutes pre/post MAST)	11 (6.0)	26 (5.7)	21 (4.7)	0.04
Salivary cortisone				
AUC _t1t7_ (nmol/L*min)	148 (56)	193 (78)	161 (70)	0.10
C_max_ (nmol/L)	32 (4.0)	32 (5.2)	31 (4.5)	0.52
T_max_ (minutes pre/post MAST)	15 (5.8)	31 (6.4)	24 (4.5)	0.04

Data are expressed as mean (SEM). AUC_t1t7,_ area under the curve from 15 min pre-MAST to 75 min post-MAST; Cmax, maximum concentration; Tmax, time to reach Cmax; *p*-value for treatment effect derived from the Friedman’s test.

## Data Availability

The data presented in this study are available on request from the corresponding author.

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
