# Peer review of "Acute Effect of a Saffron Extract (Safr’InsideTM) and Its Main Volatile Compound on the Stress Response in Healthy Young Men: A Randomized, Double Blind, Placebo-Controlled, Crossover Study"

_nutrients, 2023, doi:10.3390/nu15132921_

Round 1

Reviewer 1 Report

Thank you for the opportunity to peer review your paper looking at the acute effects of a saffron extract on stress response in healthy young men.

I commend you on taking a stab at stress response, your methodology was took into account how many variables can come into play when trying to measure stress. As an initial proof of concept randomized controlled trial  (RCT) you tried to minimize the confounding factors as best you could. I appreciate the use of visual analogue scales (VAS) to establish the psychological component of anxiety and stress perception and the measurement of cortisol/cortisone in saliva to establish the biological reponse. That being said, I believe your introduction could be strengthen by referencing the studies that show  the safety and metabolic data available for the trade-marked product used, I would recommend referencing the RCT that exist for mood as this adds credence to your hypothesis as the interplay between anxiety and depression has been extensively studied and proven throughout the medical literature, the the existing data for anxiety presented is faulty (as you point out).

If possible I would also recommend a larger or clearer image of your CONSORT diagram. It is essential to understanding the methods, and as currently presented it is impossible to read (I had to go to the digital version and zoom in quite a bit).

It would also be useful to have a visual timeline of when the assessments/ samples were done to insure reproducibility of your methods which I found cumbersome to follow and could probably be helped by some editing and rewording.

I look forward to following the evidence for this compound.

The methods section should be edited to improve the flow and reader's understanding of what was done. I would also recommend reading it through as there are some obvious typos (ie. line 69, 100,199, 201 and 202) and some non-native English-speaker sentence structures that are cumbersome (ie. line 46, 134, 144-147, 151, etc.)

Author Response

Thank you for the opportunity to peer review your paper looking at the acute effects of a saffron extract on stress response in healthy young men.

I commend you on taking a stab at stress response, your methodology was took into account how many variables can come into play when trying to measure stress. As an initial proof of concept randomized controlled trial (RCT) you tried to minimize the confounding factors as best you could. I appreciate the use of visual analogue scales (VAS) to establish the psychological component of anxiety and stress perception and the measurement of cortisol/cortisone in saliva to establish the biological response.

The authors are grateful for the valuable comments and suggestions from the Editor and Reviewers. We have taken all comments into consideration and have revised the manuscript accordingly.

That being said, I believe your introduction could be strengthen by referencing the studies that show the safety and metabolic data available for the trade-marked product used, I would recommend referencing the RCT that exist for mood as this adds credence to your hypothesis as the interplay between anxiety and depression has been extensively studied and proven throughout the medical literature, the existing data for anxiety presented is faulty (as you point out).

The randomized controlled trial published about Safr’Inside has been cited in the initial version of the manuscript as the reference 20. However, the reviewer is right, we did not emphasize neither his brand name nor his safety nor his efficacy regarding low mood. As suggested, we have modified the introduction as well : The second clinical study showed a beneficial effect of a single dose of Safr’InsideTM on another biological response to stress, the heart rate variability (Root Mean Square of the Successive Differences (RMSSD)), after an induced laboratory acute stress (the Observed Multitasking Stressor) in healthy adults experiencing low mood [22]. Indeed, participants who received placebo had a decreased RMSSD during the laboratory stressor whereas participants receiving Safr’InsideTM experienced no change in RMSSD (p=0.003). Although the effect of saffron on the HPA axis was assessed, the participants had anticipated the stressor rendering the results uninterpretable. Otherwise, Safr’Inside was well tolerated and associated with reduced depression scores and improved social relationships after 8 weeks supplementation.

If possible I would also recommend a larger or clearer image of your CONSORT diagram. It is essential to understanding the methods, and as currently presented it is impossible to read (I had to go to the digital version and zoom in quite a bit).

The CONSORT diagram has been modified in order to improve the understanding of the method.

It would also be useful to have a visual timeline of when the assessments/ samples were done to insure reproducibility of your methods which I found cumbersome to follow and could probably be helped by some editing and rewording.

We have added a figure describing the testing timeline of the clinical visits. As the maximum number of tables and figures was reached, we had this new figure as the supplementary figure S1 at the end of the manuscript.

I look forward to following the evidence for this compound.

Thank you very much for this positive comment.

Comments on the Quality of English Language

The methods section should be edited to improve the flow and reader's understanding of what was done. I would also recommend reading it through as there are some obvious typos (ie. line 69, 100,199, 201 and 202) and some non-native English-speaker sentence structures that are cumbersome (ie. line 46, 134, 144-147, 151, etc.)

We apologized for these As recommended by the reviewer, we have corrected the typos and changed the structure of some sentences judged cumbersome in the method section (particularly in the study design and protocol) in order to improve the reader’s understanding.

Reviewer 2 Report

I would appreciate the opportunity to review the article. Congratulations on the work and the detail with which you have written it. After some reformulation, I consider it ready for publication.

Lina 46: what do the authors mean by today's modern lifestyle?

Line 56: suggested to further explore this relationship between anxiolytics and other alternatives in stress, for example with meta analysis studies.

Line 61-76: it would be interesting to present some statistical results. 

Regarding the exclusion criteria, why did the authors exclude several potential beneficiaries from the study?

The work is written in detail. 

Line 184: The authors do not mention why they use an effect size of 1.4.

Why do the authors talk about population characteristics in the results section?

"Firstly, as greater acute HPA responses have been found in adult men as compared to adult women [38], we investigated the stress response only in healthy young men to increase the likelihood to detect an anti-stress effect of the tested product," To what extent isn't this biasing your results?

Author Response

I would appreciate the opportunity to review the article. Congratulations on the work and the detail with which you have written it. After some reformulation, I consider it ready for publication.

The authors are grateful for the valuable comments and suggestions from the Editor and Reviewers. We have taken all comments into consideration and have revised the manuscript accordingly.

Lina 46: what do the authors mean by today's modern lifestyle?

« Today’s modern lifestyle » means stressful lifestyle, especially for workers in developed countries that are prone to multiple sources of stress.  We have clarified this point:

In our modern life, people may be daily exposed to different source of stress such as constant digital connection, heavy workload, changes in sleep wake schedules or fear of climate change.

Line 56: suggested to further explore this relationship between anxiolytics and other alternatives in stress, for example with meta-analysis studies.

Thank you for this suggestion. We have added two meta-analyses to further explore relationship between anxiolytics and other alternatives in stress in the introduction:

Combining pharmacological and non-pharmacological interventions such as nutraceuticals, phytoceuticals or psychological interventions to manage stress-related problems may be the most effective approach taking into account individual patients’ characteristics [1,2].    

Line 61-76: it would be interesting to present some statistical results. 

Some statistical results of two clinical studies (references 19 and 20) were added:

The first one has shown that female affected by menstrual distress and who had breathed saffron for 20 minutes experienced significant decreased salivary cortisol coupled with reduced self-perceived anxiety compared to those receiving placebo, suggesting a potential anti-stress and anxiolytic effect likely induced by the volatile compound safranal [3]. Indeed, means of salivary cortisol was about 0.11 ug/dl after saffron breathing compared to 0.18 ug/dl in the placebo group (p<0.0051). The second clinical study showed a beneficial effect of a single dose of Safr’InsideTM on another biological response to stress, the heart rate variability (Root Mean Square of the Successive Differences (RMSSD)), after an induced laboratory acute stressor (the Observed Multitasking Stressor) in healthy adults experiencing low mood [4]. Indeed, participants who received placebo had a decreased RMSSD during the laboratory stressor whereas participants receiving Safr’InsideTM experienced no change in RMSSD (p=0.003). Although the effect of saffron on the HPA axis was assessed, the participants had anticipated the stressor rendering the results uninterpretable. Otherwise, Safr’Inside was well tolerated and associated with reduced depression scores and improved social relationships after 8 weeks supplementation.

Regarding the exclusion criteria, why did the authors exclude several potential beneficiaries from the study?

The objective of this study was to investigate the acute effect of Safr’InsideTM and its volatile compound safranal on the biological and psychological stress responses in healthy men. Safr’InsideTM is a standardized saffron extract that can be included in food supplements. According to the European Food Safety Autority [5], food supplements are intended to supplement the normal diet, or to support specific physiological functions. They are not medicinal products and their use is not intended to treat or prevent diseases in humans and their efficacy should be tested in healthy populations.

  • As the study population must be conducted in healthy subjects, participants having any acute or chronic disease were considered as unhealthy and thus were ineligible for this study. Thus, participants with high blood pressure, history or currently suffering from mental disorders, infectious diseases, diabetes mellitus, allergic diseases, cardiovascular diseases, unbalance thyroid diseases were excluded to limit selection bias.
  • Variability in cortisol reactivity is influenced by both genetic and environmental factors [6] such as stressful events, change in sleeping habits, biological sex, alcohol, foods or dietary supplements, coffee, tobacco, steroids, etc. All these criteria are confusion bias that may potentially influence the cortisol secretion and consequently bias the interpretation of the results.
  • As greater acute HPA responses have been found in adult men as compared to adult women [7], we investigated the stress response only in healthy young men in order to increase the likelihood to see a pic of cortisol in response to stress and to detect an anti-stress effect of the tested products. In addition, the influence of endogenous or exogenous sex hormones are known to modulate the cortisol levels in blood. That’s why women were excluded from the study population.

Finally, these exclusion criteria allow to enroll a homogenous population to reduce variability of cortisol response to the stressor.

The work is written in detail. 

Line 184: The authors do not mention why they use an effect size of 1.4.

We apologized for this misunderstanding. Effect size of 1.4 was based on the means and SD of salivary cortisol from the figure 5 of Scholey and al.[8] 1 hour post-treatment. We have clarified this point in the manuscript:

As no clinical study has investigated the effect of saffron extract in response to an acute laboratory stressor on salivary cortisol response, the sample size calculation was based on the effect size of a lemon balm food supplement on salivary cortisol in healthy adults 1 hour post-treatment [9]. A sample size of 18 subjects was hypothesized to detect an effect size of 1.4 on salivary cortisol response, with a two-sided 5% significance level and a power of 80%.

 Why do the authors talk about population characteristics in the results section?

The population characteristics are described in the results section as recommended by the Consolidated Standards of Reporting Trials (CONSORT) statement developed to improve the reporting of randomized controlled trials [10]. Baseline demographic and clinical characteristics of the study population should be tabulated in the results section. This description allows to the reader to check that the focused population was indeed enrolled without major deviation.

"Firstly, as greater acute HPA responses have been found in adult men as compared to adult women [38], we investigated the stress response only in healthy young men to increase the likelihood to detect an anti-stress effect of the tested product," To what extent isn't this biasing your results?

We agree that our study population was restricted to healthy young men and cannot be generalized to women or older population, that’s why we mentioned this point in the limitations.

References

  1. Coventry, P.A.; Meader, N.; Melton, H.; Temple, M.; Dale, H.; Wright, K.; Cloitre, M.; Karatzias, T.; Bisson, J.; Roberts, N.P.; et al. Psychological and Pharmacological Interventions for Posttraumatic Stress Disorder and Comorbid Mental Health Problems Following Complex Traumatic Events: Systematic Review and Component Network Meta-Analysis. PLoS Med. 2020, 17, doi:10.1371/JOURNAL.PMED.1003262.
  2. Sarris, J.; Ravindran, A.; Yatham, L.N.; Marx, W.; Rucklidge, J.J.; McIntyre, R.S.; Akhondzadeh, S.; Benedetti, F.; Caneo, C.; Cramer, H.; et al. Clinician Guidelines for the Treatment of Psychiatric Disorders with Nutraceuticals and Phytoceuticals: The World Federation of Societies of Biological Psychiatry (WFSBP) and Canadian Network for Mood and Anxiety Treatments (CANMAT) Taskforce. World J. Biol. Psychiatry 2022, 23, 424–455, doi:10.1080/15622975.2021.2013041.
  3. Fukui, H.; Toyoshima, K.; Komaki, R. Psychological and Neuroendocrinological Effects of Odor of Saffron (Crocus Sativus). Phytomedicine 2011, 18, 726–730, doi:10.1016/J.PHYMED.2010.11.013.
  4. Jackson, P.A.; Forster, J.; Khan, J.; Pouchieu, C.; Dubreuil, S.; Gaudout, D.; Moras, B.; Pourtau, L.; Joffre, F.; Vaysse, C.; et al. Effects of Saffron Extract Supplementation on Mood, Well-Being, and Response to a Psychosocial Stressor in Healthy Adults: A Randomized, Double-Blind, Parallel Group, Clinical Trial. Front. Nutr. 2021, 7, doi:10.3389/fnut.2020.606124.
  5. DIRECTIVE 2002/46/EC OF THE EUROPEAN PARLIAMENT AND OF THE COUNCIL of 10 June 2002 on the Approximation of the Laws of the Member States Relating to Food Supplements (Text with EEA Relevance).
  6. Khoury, J.E.; Gonzalez, A.; Levitan, R.D.; Pruessner, J.C.; Chopra, K.; Basile, V.S.; Masellis, M.; Goodwill, A.; Atkinson, L. Summary Cortisol Reactivity Indicators: Interrelations and Meaning. Neurobiol. Stress 2015, 2, 34–43, doi:10.1016/J.YNSTR.2015.04.002.
  7. Verma, R.; Balhara, Y.P.S.; Gupta, C.S. Gender Differences in Stress Response: Role of Developmental and Biological Determinants. Ind. Psychiatry J. 2011, 20, 4, doi:10.4103/0972-6748.98407.
  8. Scholey, A.; Gibbs, A.; Neale, C.; Perry, N.; Ossoukhova, A.; Bilog, V.; Kras, M.; Scholz, C.; Sass, M.; Buchwald-Werner, S. Anti-Stress Effects of Lemon Balm-Containing Foods. Nutrients 2014, 6, 4805–4821, doi:10.3390/nu6114805.
  9. Scholey, A.; Gibbs, A.; Neale, C.; Perry, N.; Ossoukhova, A.; Bilog, V.; Kras, M.; Scholz, C.; Sass, M.; Buchwald-Werner, S. Anti-Stress Effects of Lemon Balm-Containing Foods. Nutrients 2014, 6, 4805–4821, doi:10.3390/NU6114805.
  10. Dwan, K.; Li, T.; Altman, D.G.; Elbourne, D. CONSORT 2010 Statement: Extension to Randomised Crossover Trials. BMJ 2019, 366, doi:10.1136/BMJ.L4378.

Round 2

Reviewer 1 Report

Thank you for your thoughtful reviews; the manuscript now reads cohesively throughout. Thank you for adding a higher resolution CONSORT and for the supplementary figure, I believe that the soundness of your methodology is better served with these changes. Thank you for the editing done to the introduction as it provides credence to your hypothesis.

A few minor editing changes needed, but overall in great shape.